molecular biology/genetics/genomics

Mexican tetra, gluconeogenesis, nutrition, brain, liver

**Author for correspondence:**
Lucie Marandel
e-mail: lucie.marandel@inrae.fr

# Nutritional regulation of glucose metabolism-related genes in the emerging teleost model Mexican tetra surface fish: a first exploration

Lucie Marandel[1], Elisabeth Plagnes-Juan[1],
Michael Marchand[1], Therese Callet[1], Karine Dias[1],
Frederic Terrier[1], Stéphane Père[2], Louise Vernier[2],
Stephane Panserat[1] and Sylvie Rétaux[2]

[1]INRAE, Université de Pau & Pays de l'Adour, E2S UPPA, UMR1419 Nutrition Metabolism and Aquaculture, Aquapôle, 64310 Saint-Pée-sur-Nivelle, France
[2]Paris-Saclay Institute of Neuroscience, CNRS UMR9197, Université Paris-Saclay, Avenue de la terrasse, Gif-sur-Yvette, France

LM, 0000-0001-9395-6041

*Astyanax mexicanus* has gained importance as a laboratory model organism for evolutionary biology. However, little is known about its intermediary metabolism, and feeding regimes remain variable between laboratories holding this species. We thus aimed to evaluate the intermediary metabolism response to nutritional status and to low (NC) or high (HC) carbohydrate diets in various organs of the surface-dwelling form of the species. As expected, glycaemia increased after feeding. Fish fed the HC diet had higher glycaemia than fish fed the NC diet, but without displaying hyperglycaemia, suggesting that carbohydrates are efficiently used as an energy source. At molecular level, only *fasn* (*Fatty Acid Synthase*) transcripts increased in tissues after refeeding, suggesting an activation of lipogenesis. On the other hand, we monitored only moderate changes in glucose-related transcripts. Most changes observed were related to the nutritional status, but not to the NC versus HC diet. Such a metabolic pattern is suggestive of an omnivorous-related metabolism, and this species, at least at adult stage, may adapt to a fish meal-substituted diet with high carbohydrate content and low protein supply. Investigation

to identify molecular actors explaining the efficient use of such a diet should be pursued to deepen our knowledge on this species.

# 1. Introduction

*Astyanax mexicanus (A. mexicanus)*, a freshwater characiform teleost, is a single species with two different forms, a river-dwelling (surface fish) and a cave-adapted (cavefish) morphotype. The latter form is considered as a natural mutant of the ancestral surface-like form with divergent phenotypic traits [1]. Because of this particularity, *A. mexicanus* has emerged as a master natural model for investigations in the field of evolutionary biology but not only [2]. Indeed, this species, owing to its biology, is easy to maintain (small size) and propagate (large number of eggs and relatively short generation time) in laboratory conditions [3]. Moreover, useful tools to perform functional genomic analyses are now validated in this species [3–5]. These advances promote *A. mexicanus* to the rank of a laboratory model organism in the same way as the zebrafish or the medaka. For these reasons, trying to improve breeding conditions of this fish continues to raise attention. One point, which has never been explored, is the nutritional requirement of the surface *Astyanax* with respect to its intermediary metabolism response. This is particularly important in adult fish, which are kept many years in laboratories as breeding colonies. Actually, surface fish is usually described as highly carnivorous, feeding on smaller fish and invertebrates. However, its stomach can sometimes contain plant-stuff, hence a better categorization would probably be omnivorous (Fishbase: https://www.fishbase.in/summary/Astyanax-mexicanus.html). Despite these data collected in the field, most of the surface *Astyanax* are fed on a carnivorous-based formula diet in captivity [3,6,7] containing between 38 and 59% of proteins, the major part of which coming from fish meal. In omnivorous fish, part of fish meal is commonly substituted by dietary carbohydrates, a less expensive raw material for aquafeed formula. However, up to now, nothing is known about the regulation of the intermediary metabolism of the surface *A. mexicanus*. Acquiring knowledge on fundamental nutritional metabolism pathway related to glucose use is essential before exploring the nutritional capability of this fish to use digestible carbohydrates. We thus addressed this question in the present study by testing the glucose and lipid metabolism of surface *A. mexicanus* fed either a no carbohydrate diet or a diet containing 30% of digestible carbohydrates, which is considered very high for carnivorous-related metabolism or medium for omnivorous-related metabolism.

In the last decade, several studies have highlighted the existence of nutrient sensing systems in fish (recently reviewed by Conde-Sieira & Soengas [8]). In particular, glucosensing capacity has been identified at both central (hypothalamus, hindbrain and more recently in the telencephalon of the carnivorous trout [9]) and peripheral (liver and intestine) locations. The present study thus focused on organs known to have major roles in glucose homeostasis/sensing: intestine, the gateway to the body for glucose through glucose transporters; liver, which has a central role in the regulation of blood glucose in the post-absorptive state; muscle, because its mass and contractile activity represents a major captor of glucose for energy purpose; and brain, involved in nutrient sensing. Our attention was mainly focused on well-known actors of glucose metabolism whose atypical regulation at molecular level (i.e. mRNA level) is linked to the poor capacity of using dietary carbohydrates in fish [10,11]. This included glucose transporters; and enzymes involved in glycolysis, the main metabolic pathway for glucose assimilation and energy production and its reverse process, gluconeogenesis, the pathway of endogenous glucose production from pyruvate and gluconeogenic amino acids. We also analysed the first step of pentose phosphate production, a major alternate pathway by which glucose can be broken down to produce reducing power considered as the biological energy currency. As glucose calories can be stored by the conversion of acetyl-coA (a central metabolite of glycolysis/pentose phosphate) to fatty acids, fatty acids synthase was also analysed at molecular level. Lastly, as the response of intermediary metabolism was never tested before in the surface fish regarding its nutritional status, our analysis was also conducted in fasted fish.

# 2. Material and methods

## 2.1. Fish, ethical issues and approval

Laboratory stocks of *A. mexicanus* surface fish (originating from San Solomon Spring, Balmorhea State Park, TX, USA) were obtained in 2004 from the Jeffery laboratory at the University of Maryland, College Park,

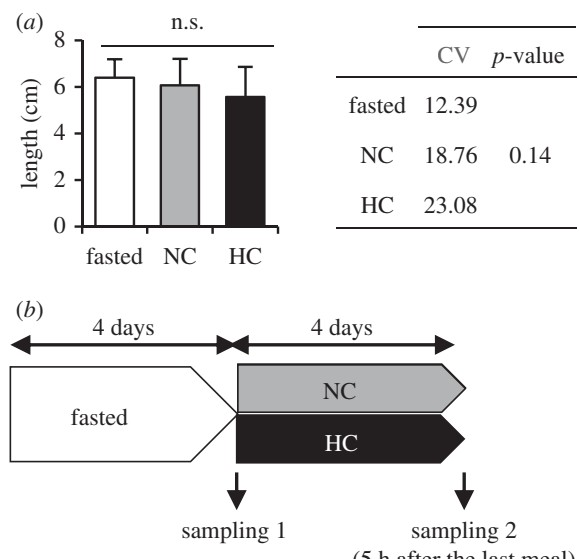

**Figure 1.** Zootechnical parameters (length) of sampled fish (*a*) and experimental design (*b*). Representative blot results are given for Akt analysis in muscle (E). NC, no carbohydrate diet; HC, high carbohydrate diet; CV, coefficient of variation. Data are presented as mean ± s.d. values of *n* = 12 fish per condition except for HC condition where *n* = 11.

MD, USA. Since then, in our facility, fish were bred and maintained at 25–26°C on a 12 : 12 h light : dark cycle [3]. Investigations were conducted according to the guiding principles for the use and care of laboratory animals and in compliance with French and European regulations on animal welfare (Décret 2001-464, 29 May 2001, and Directive 2010/63/EU, respectively). S.R.'s authorization for use of animals in research including *Astyanax mexicanus* is 91-116. The Paris Centre-Sud Ethic Committee approved the protocol authorization number 2017-04#8545 related to the present research.

## 2.2. Diets and experimental design

Adult surface *A. mexicanus* born in our facility and aged between 2 and 10 years were habituated for one week in groups of 12 at 25°C, in duplicates of tanks per type of diet. Fish were split to respect an equivalent distribution in terms of length (correlated to the age of fish as previously described [12]) between tanks/conditions. This was confirmed by no statistical difference between the coefficient of variation of the three conditions (figure 1*a*). The first sampling was performed after 4 days of fasting. Fish were then fed with either the NC (containing no carbohydrate) or the HC diet (containing 30% carbohydrates) twice a day at 2.5% live weight for 4 days and sampled 5 h after the last meal (figure 1*b*). Diet compositions are provided in table 1 and their proximal composition was analysed as previously described [13]. Gut content of sacrificed animals was systematically checked to confirm that the fish had consumed the food. Fresh livers, muscle, intestine and brain (divided into three parts containing hypothalamus, hindbrain and telencephalon, respectively) of 12 fish per condition were dissected, immediately frozen in liquid nitrogen and then kept at −80°C until analyses. Six fish were used for molecular analysis and six for western blot analysis.

## 2.3. Blood glucose analysis

Glycaemia was measured using FreeStyle Lite blood glucose meter (Abbott) after having collected blood from the heart directly on the test strip during dissection.

## 2.4. Western blot analysis

Our study expanded upon the methods initially described in [14]. Frozen muscles (10–100 mg) were weighted into 1 ml of lysis buffer (150 mM NaCl, 10 mM Tris, 1 mM EGTA, 1 mM EDTA (pH 7.4), 100 mM sodium fluoride, 4 mM sodium pyrophosphate, 2 mM sodium orthovanadate, 1% Triton X-100, 0.5% NP-40-IGEPAL, and a protease inhibitor cocktail (Roche, Basel, Switzerland)) and homogenized using Precellys-Cryolys homogenizer Bertin (6500 r.p.m., three times, 20 s on, 20 s off). Homogenates

**Table 1.** Formulation and proximate composition of the two experimental diets used (NC and HC diets) in this experiment.

| | NC | HC |
|---|---|---|
| fish meal[a] | 83.15 | 51.17 |
| fish oil[b] | 12.85 | 14.83 |
| corn pregelatinized starch[c] | 0 | 30 |
| vitamin mix[d] | 1 | 1 |
| mineral mix[e] | 1 | 1 |
| alginate[f] | 2 | 2 |
| *proximate composition* | | |
| dry matter (MS) | 96.9 | 96.0 |
| proteins (% DM) | 62.9 | 39.2 |
| lipids (% DM) | 19.6 | 18.8 |
| energy (% DM) | 23.0 | 22.3 |
| ash (% DM) | 15.3 | 10.2 |
| carbohydrate (% DM) | <1 | 26.1 |

[a]Sopropeche, Boulogne-sur-Mer, France.

[b]fish oil; Sopropeche, Boulogne-sur-Mer, France.

[c]Gelatinized corn starch; Roquette, Lestrem, France.

[d]Supplied the following (/kg diet): DL-a tocopherol acetate 60 IU, sodium menadione bisulfate 5 mg, retinyl acetate 15 000 IU, DLcholecalciferol 3000 IU, thiamin 15 mg, riboflavin 30 mg, pyridoxine 15 mg, vit. B12 0.05 mg, nicotinic acid 175 mg, folic acid 500 mg, inositol 1000 mg, biotin 2.5 mg, calcium panthotenate 50 mg, choline chloride 2000 mg.

[e]Supplied the following (/kg diet): calcium carbonate (9.40% Ca) 2.15 g, magnesium oxide (60% Mg) 1.24 g, ferric citrate 0.2 g, potassium iodide (75% I) 0.4 g, zinc sulfate (36% Zn) 0.4 g, copper sulfate (25% Cu) 0.3 g, manganese sulfate (33% Mn) 0.3 g, dibasic calcium phosphate (20% Ca, 18% P) 5 g, cobalt sulfate 2 mg, sodium selenite (30% Se) 3 mg, potassium chloride 0.9 g, sodium chloride 0.4 g.

[f]Louis François, Marne-la-Vallée, France.

were centrifuged at 1000$g$ for 15 min at 4°C, and we recovered the supernatant to centrifuge again at 20 000$g$ for 30 min at 4°C. The resulting supernatant fractions were obtained and stored at −80°C. Protein concentrations were determined using the BCA protein assay. Lysates (23.5 µg of the total protein) were subjected to SDS–PAGE. Appropriate antibodies were obtained from Cell Signaling Technologies (Ozyme, saint Quentin Yvelines, France). Anti-phospho-AKT (Ser473) (no. 4060) and anti-AKT (no. 9272) were used for western blot analysis. All the antibodies have been verified to cross-react successfully with the surface Mexican fish. After washing, membranes were incubated with an IRDye infrared secondary antibody (LI-COR Biosciences). The bands were visualized by infrared fluorescence using the Odyssey imaging system (LI-COR Biosciences) and quantified by Odyssey infrared imaging system software (v. 3.0, LI-COR biosciences).

## 2.5. mRNA levels analysis

Our study expanded upon the methods initially described in [15]. Total RNA samples were analysed on the liver. Samples were extracted using TRIzol reagent (Invitrogen), according to the manufacturer's recommendations and were quantified by spectrophotometry (absorbance at 260 nm). The integrity of the samples was assessed using agarose gel electrophoresis. One microgram of total RNA per sample was reverse transcribed into cDNA using the SuperScript III reverse transcriptase kit (Invitrogen) with random primers (Promega, Charbonnieres, France) according to the manufacturer's instructions.

mRNA levels of glucose and lipid metabolism-related genes were determined by quantitative real-time (q) RT–PCR. Primer sequences are given in electronic supplementary material, table S1. Elongation factor-1 α (ef1α) for muscle, 18S for the liver and β-actin for the intestine and brain (all parts) were considered as reference genes. Each PCR product was systematically sequenced.

Quantitative RT–PCRs were carried out on a Light Cycle 480 II (Roche Diagnostics, Neuilly-sur-Seine, France) using SYBR Green I Master (Roche Diagnostics GmbH, Mannheim, Germany). PCR were

performed using 2 µl of the diluted cDNA (76 times) mixed with 0.24 µl of each primer (10 µM), 3 µl of Light Cycle 480 SYBR Green I Master (Roche Diagnostics) and 0.52 µl of DNase/RNase/protease-free water (5 prime, Hamburg, Germany) in a total volume of 6 µl. The qPCR were initiated at 95°C for 10 min, then followed by 45 cycles of a three-step amplification programme (15 s at 95°C, 10 s at 60°C, 15 s at 72°C). Melting curves were systematically monitored (5 s at 95°C, 1 min at 65°C, temperature gradient 0.11°C s$^{-1}$ from 65 to 97°C) at the end of the last amplification cycle to confirm the specificity of the amplification reaction. Each PCR assay included replicate samples (duplicate of reverse transcription and PCR amplification, respectively) and negative controls (reverse transcriptase and RNA free samples). Relative quantification of target genes expression was performed using the E-Method from the Light Cycler 480 software (v. SW 1.5; Roche Diagnostics). PCR efficiencies were measured by the slope of a standard curve using serial dilution of cDNA, and they ranged between 1.90 and 2.0.

## 2.6. Statistical analysis

The normality of distributions was assessed using the Shapiro–Wilk test. Data were then analysed by a Kruskal–Wallis non-parametric test followed by a Tukey test as a *post hoc* analysis. All analyses were performed using R Commander package in R (v. 3.1.0, https://cran.r-project.org/web/packages/Rcmdr/).

Coefficients of variation (CV) were calculated as the ratio between standard deviation and mean. Differences between CV were evaluated by a $\chi^2$ test using R software.

# 3. Results and discussion

The present study aimed at exploring the glucose metabolism response of the newly emerging laboratory model, the surface *A. mexicanus*, regarding its nutritional status (i.e. fasted versus fed) but also when challenged with a diet containing a high content of digestible carbohydrates.

## 3.1. Fish ate well during the nutritional challenge

During the 4 day nutritional challenge, fish were fed twice a day at 2.5% live weight with either the NC or the HC diet. As fish sizes were heterogeneous (from 3.9 to 8 cm), the granulometry of diets (diameter 1 mm) was carefully chosen. Pellet size thus ranged between 800 and 1200 µm, these diameters being determined to allow all fish eating based on what is commonly recommended for trout alevins.

The first step of our investigation was to confirm that fish ate well during the nutritional challenge. To do so, we analysed the phosphorylation of Akt (p-Akt), a biomarker of nutritional status, and the blood glucose level. The analysis of p-Akt was conducted in muscle instead of the liver because of our sampling timing (5 h post-prandial), as this signalling pathway is activated first in the liver and then in the muscles. Our results demonstrated statistically significant differences in p-Akt/Akt ratio between experimental conditions, with the fed fish showing higher levels of Akt phosphorylation than the starved fish, as expected (figure 2a).

Concerning glycaemia, our results showed that fed fish displayed a higher glycaemia than fasted fish (figure 2b). These results confirmed that fish ate well during the experiment. Interestingly, blood glucose levels of fasted fish were surprisingly low compared to data obtained in other teleosts (carnivorous trout [15] or omnivorous tilapia [16,17], zebrafish [18] or goldfish [19]) but were in accordance with results previously published by Riddle et al. [6] and measured with the same glucometer reference.

## 3.2. Plasmatic glycaemia is in favour of an efficient use of dietary carbohydrates

The second step of our investigation was to assess the effect of a modification of the protein–carbohydrate ratio in the diet on the previous parameters (i.e. Akt phosphorylation and glycaemia).

The p-Akt/Akt ratio was overall significantly different between our experimental conditions, but the *post hoc* test was not significant (figure 2a). p-Akt/Akt slightly differed between fish fed the NC diet and fish fed the HC diet. This finding was in accordance with previous data obtained in zebrafish, an omnivorous fish, when fed a high carbohydrate diet [14].

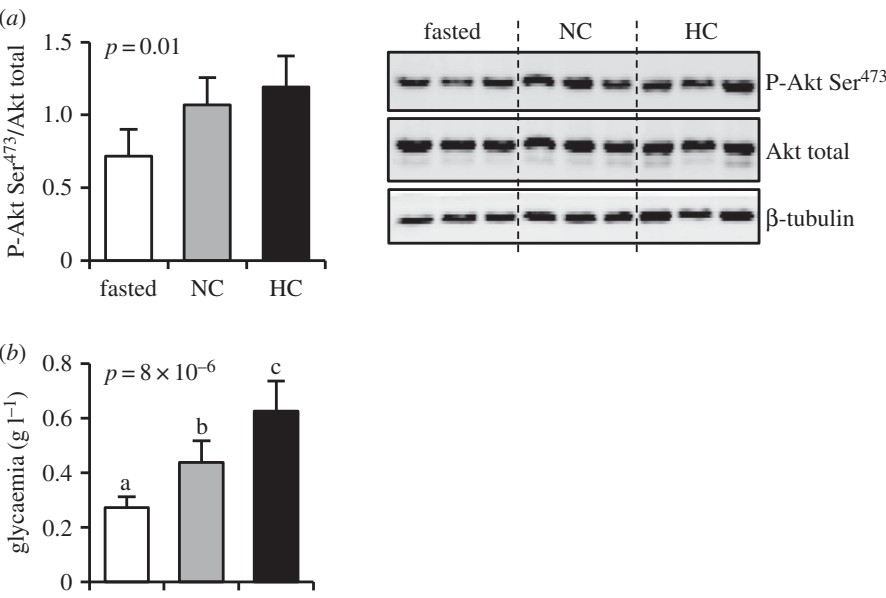

**Figure 2.** Muscular Akt phosphorylation (*a*) and glycaemia (*b*) of sampled fish. Representative blot results are given for Akt analysis. NC, no carbohydrate diet; HC, high carbohydrate diet; CV, coefficient of variation. Data are presented as mean ± s.d. values of $n = 6$ fish except for NC condition where $n = 5$.

Regarding glycaemia, blood glucose level is closely linked to the ingestion/digestion of digestible carbohydrates. Fish fed the HC diet had a significantly higher glycaemia than fish fed the NC diet (figure 2*b*), confirming that *A. mexicanus* ingested and digested dietary carbohydrates. However, the fold-change of glycaemia in HC-fed fish with respect to NC-fed fish is around 1.3. Of note, in carnivorous fish which are poor users of dietary carbohydrates, such as trout, this fold-change reaches 3–4 [15,20]. The relatively weak increase in blood glucose level in *A. mexicanus* after HC diet may reflect an efficient use of digestible carbohydrates, as previously described in omnivorous fish such as the Nile tilapia [17], but in contrast with the situation in carnivorous fish such as the rainbow trout [15]. However, as for Nile tilapia, the fold-change of glycaemia between NC and HC-fed fish is lower than 2, we cannot consider that fish are hyperglycaemic in the present study.

The ability to modulate their intermediary metabolism could be one of the main hypotheses to explain the phenotypic response of the surface Mexican fish to a high carbohydrate diet.

## 3.3. Glucose and lipid metabolism are more affected by the nutritional status than by the dietary carbohydrate content at molecular level

As indicated in the Introduction, several tissues play a critical role in glucose homeostasis and more generally in dietary nutrient assimilation and catabolism. Specific actors of glucose metabolism (glycolysis, gluconeogenesis, glucose transporter-related genes etc.) are known to be modulated at molecular level in these tissues, and, when atypically regulated, they are proposed to be linked to the poor use of dietary carbohydrates in some species (for instance, in trout [15,21]). Therefore, we analysed these actors to evaluate their regulation by nutritional status as well as by different diets. Correlations between mRNA levels and the length of fish (correlated to the age of the fish, see §3.2) were also tested (data not shown) as fish had variable ages. No statistically significant correlation was found, suggesting that, in our experiment and for the tested genes, the age of the fish did not influence the results.

### 3.3.1. Intestine: glucose uptake

Once a diet is digested, nutrients enter the body by firstly passing through the intestinal barrier using transporters—i.e. glucose transporters when considering this nutrient [22]. In the intestine, *glut1a* mRNA level were lower in fed fish than in fasted fish (table 2), whereas *glut3b* tended to decrease in

**Table 2.** mRNA levels of glucose metabolism-related genes in liver, muscle and intestine of fasted fish (F) or fed the no (NC) or high (HC) carbohydrate diet. Data are presented as mean ± s.d. value ($n = 6$ fish per condition). A Kruskal–Wallis non-parametric test was performed between all conditions ($p$-values are indicated in the table) and when significant ($p < 0.05$, lines in bold), it was followed by a Tukey test as a *post hoc* analysis. Different letters stand for significant differences between conditions. Additional Kruskal–Wallis non-parametric test was performed between NC and HC conditions ($p$-values not shown).

| genes | liver | | | | muscle | | | | intestine | | | |
|---|---|---|---|---|---|---|---|---|---|---|---|---|
| | F | NC | HC | p-value | F | NC | HC | p-value | F | NC | HC | p-value |
| *gluconeogenesis* | | | | | | | | | | | | |
| g6pca | 1.8 ± 0.9 | 0.4 ± 0.2 | 0.6 ± 0.5 | 0.17 | | | | | not detected | | | |
| g6pcb1 | 1.4 ± 0.7 | 0.6 ± 0.5 | 1.0 ± 0.6 | 0.13 | | | | | 4.2 ± 5.2 | 0.7 ± 0.3 | 1.4 ± 1.6 | 0.42 |
| g6pcb2 | | not detected | | | | | | | not detected | | | |
| fbp1a | 0.7 ± 0.4 | 1.1 ± 0.9 | 1.3 ± 1.0 | 0.75 | | | | | 1.3 ± 0.7 | 1.0 ± 0.3 | 1.1 ± 0.7 | 0.71 |
| fbp1b | 1.4 ± 0.9 | 0.7 ± 0.5 | 1.1 ± 0.6 | 0.12 | | | | | 4.9 ± 6.4 | 1.4 ± 0.5 | 2.7 ± 2.0 | 0.23 |
| fbp2 | | | | | | | | | | | | |
| pck1 | 1.7 ± 1.2 | 0.9 ± 1.1 | 0.2 ± 0.2 | 0.05 | | | | | not detected | | | |
| pck2 | 0.7 ± 0.2 | 1.1 ± 1.0 | 1.2 ± 1.0 | 0.53 | | | | | 1.7 ± 1.7 | 1.1 ± 0.3 | 1.4 ± 0.7 | 0.69 |
| *glycolysis* | | | | | | | | | | | | |
| gck | 0.1 ± 0.1[a] | 1.7 ± 2.3[b] | 1.1 ± 1.2[b] | **0.02** | | | | | 1.6 ± 2.7 | 0.9 ± 0.9 | 1.4 ± 1.6 | 0.78 |
| hk2 | | | | | 0.9 ± 0.4 | 0.9 ± 0.5 | 1.0 ± 0.2 | 0.52 | | | | |
| pfkla | 0.8 ± 1.1 | 1.2 ± 0.8 | 0.7 ± 0.5 | 0.18 | | | | | 2.4 ± 1.8 | 1.8 ± 0.5 | 2.3 ± 0.9 | 0.69 |
| pfklb | 0.4 ± 0.2 | 1.0 ± 0.7 | 1.6 ± 1.4 | 0.20 | | | | | 1.3 ± 0.8 | 0.7 ± 0.5 | 0.9 ± 0.6 | 0.26 |
| pklr | 1.2 ± 0.4[a] | 1.0 ± 0.4[a] | 0.5 ± 0.1[b] | **0.01** | | | | | 2.1 ± 1.9 | 0.9 ± 0.2 | 1.7 ± 0.9 | 0.08 |
| pfkma | | | | | 1.1 ± 0.6 | 1.3 ± 1.2 | 1.3 ± 0.7 | 0.75 | | | | |
| pfkmb | | | | | 0.4 ± 0.4[a] | 1.9 ± 1.0[b] | 1.9 ± 1.4[a,b] | **0.01** | | | | |
| pkma | | | | | 1.2 ± 0.5 | 1.3 ± 0.5 | 1.0 ± 0.5 | 0.49 | | | | |
| pkmb | | | | | 0.9 ± 0.7 | 1.2 ± 0.6 | 1.7 ± 1.2 | 0.48 | | | | |

(*Continued.*)

**Table 2.** (*Continued.*)

| genes | liver | | | | muscle | | | | intestine | | | |
|---|---|---|---|---|---|---|---|---|---|---|---|---|
| | F | NC | HC | p-value | F | NC | HC | p-value | F | NC | HC | p-value |
| *transporter* | | | | | | | | | | | | |
| glut1a | not detected | | | | not detected | | | | $2.6 \pm 1.2^a$ | $0.3 \pm 0.5^b$ | $0.6 \pm 0.8^{a,b}$ | **0.01** |
| glut1b | $1.2 \pm 0.7$ | $0.9 \pm 0.8$ | $1.0 \pm 0.7$ | 0.40 | $0.9 \pm 0.8$ | $1.1 \pm 0.4$ | $1.5 \pm 1.1$ | 0.44 | $2.6 \pm 2.0$ | $0.9 \pm 0.2$ | $1.4 \pm 0.6$ | 0.07 |
| glut2 | $1.3 \pm 0.5$ | $0.4 \pm 0.1$ | $1.4 \pm 0.7^{\dagger}$ | 0.09 | | | | | $2.7 \pm 3.4$ | $0.9 \pm 0.3$ | $1.8 \pm 1.3$ | 0.17 |
| glut4 | | | | | $1.0 \pm 0.5$ | $1.0 \pm 0.5$ | $1.1 \pm 0.3$ | 0.91 | | | | |
| sglt1a | | | | | | | | | $1.6 \pm 1.2$ | $0.8 \pm 0.6$ | $2.5 \pm 1.6$ | 0.06 |
| sglt1b | | | | | | | | | $2.8 \pm 3.5$ | $0.5 \pm 0.3$ | $3.6 \pm 4.6$ | 0.05 |
| glut3a | | | | | | | | | $3.8 \pm 3.1$ | $0.8 \pm 0.3$ | $1.3 \pm 0.7$ | **0.03** |
| glut3b | | | | | | | | | $3.0 \pm 1.9$ | $0.8 \pm 0.4$ | $1.0 \pm 0.2$ | 0.05 |
| glut5 | | | | | | | | | $3.4 \pm 3.0$ | $0.6 \pm 0.5$ | $1.7 \pm 1.3$ | 0.05 |
| *lipogenesis* | | | | | | | | | | | | |
| g6pda | $1.8 \pm 1.6$ | $0.5 \pm 0.4$ | $0.8 \pm 0.5$ | **0.02** | | | | | $2.7 \pm 2.2$ | $1.0 \pm 0.4$ | $1.3 \pm 0.8$ | 0.48 |
| g6pdb | $0.1 \pm 0.0^a$ | $3.0 \pm 4.1^b$ | $0.7 \pm 1.0^b$ | **0.003** | | | | | $1.6 \pm 1.2$ | $0.9 \pm 0.5$ | $1.2 \pm 1.0$ | 0.59 |
| fasn | $0.0 \pm 0.0^a$ | $1.7 \pm 2.0^b$ | $1.6 \pm 1.6^b$ | **0.003** | | | | | $0.3 \pm 0.1$ | $1.5 \pm 1.4$ | $1.0 \pm 0.8$ | **0.03** |

$^{\dagger}$Significant differences between NC and HC.

fish fed the NC diet compared to fasted fish but increased in fish fed the HC diet (table 2), which is in accordance with a glucose uptake in this organ. In addition to its role in the uptake of glucose, the intestine can participate to the regulation of glucose homeostasis by metabolizing glucose. The glycolytic potential of the intestine has been known for a long time in several species including teleost [22–24]. In addition, a more recent gluconeogenic capacity has been highlighted for this organ, which seems to contribute to the satiety signal [25]. Here, we found that neither glycolysis nor gluconeogenesis-related transcript levels were affected by the nutritional status or the diet composition (table 2). These results are in contrast with a previous study in tilapia in which the intestine was shown to have significantly enhanced glycolytic and gluconeogenic potential in refed fish after 36 h of starvation [23]. The different results observed may be due to a question of sampling timing: 5 h after the last meal in our case, versus 3 h in the tilapia study.

### 3.3.2. Liver: the centre of metabolism

The liver is the main organ for the regulation of energy metabolism, nutrients production, storage and supply to the whole body and it has a major role in the control of glucose homeostasis. As for the intestine, glucose enters the liver using glucose transporters that we thus firstly analysed. We did not detect any expression of *glut1a*. On the other hand, *glut1b* and *glut2* were expressed but were not changed by nutritional status, whereas *glut2* increased in fish fed the HC diet (table 2). These results confirmed previous ones obtained in cod in which *glut2* was not affected by nutritional status [26]. Moreover, *glut* mRNA levels were not modified as well by dietary starch in an omnivorous cyprinid, the blunt snout bream [27]. Of note, in surface *A. mexicanus*, the mRNA levels of the two paralogous genes *glut1a* and *glut1b* thus appear to be differently regulated.

We then analysed glycolysis, which represents a universal metabolic pathway leading to glucose catabolism in all organisms [28]. We found that the mRNA levels of *gck*, which encodes glucokinase, the first enzyme of glycolysis, were increased in fed fish compared to fasted ones but were not affected by the composition of the diet (table 2). The increase in *gck* mRNA level by refeeding is similar to what is observed in the liver of mammals [29] and several teleosts species [14,30]. By contrast, in teleost, *gck* expression has been suggested to be mainly controlled by dietary carbohydrates and related to blood glucose level [30–32], contrarily to mammals in which this gene seems more regulated by insulin [33]. In common carp and perch, both omnivorous teleosts, *gck* mRNA expression is fully induced by dietary carbohydrates, meaning that the regulation by this nutrient can be species-dependent [31,32]. Overall, our data in surface *A. mexicanus* are more in favour of a mammalian-like control of *gck* transcription mostly by insulin. In addition to *gck*, the only other glycolytic gene regulated by the nutritional status, *pklr*, was downregulated in fed fish compared to fasted ones and was not enhanced by dietary carbohydrates (table 2). This gene has been previously shown to be poorly regulated by the nutritional status in zebrafish and trout [14,34,35], and fully induced by digestible dietary carbohydrates but when given in a higher proportion (50%) than in our experimental diet [14]. We can, however, not exclude that regulation occurred at post-transcriptional level.

Glycolytic carbon flux can also be diverted in cells into lipid biosynthesis by the action of Fasn (fatty acid synthase), using NADPH as a cofactor that is generated by enzymes involved in the pentose phosphate shunt, such as G6pd (glucose-6-phosphate dehydrogenase). Our results showed that genes encoding these two enzymes in the surface *A. mexicanus* were expressed in the liver and that their mRNA levels increased in refed fish. These results are in line with previous published data in zebrafish [14] and mammals [36], and suggest that this species has a good capability to synthetize lipids from dietary nutrients—an ability which seems enhanced in the cave morphotype of this species [7,37,38]. However, *fasn* and *g6pd* mRNA levels remained stable when the protein : carbohydrate ratio was decreased in the diet, which is in accordance with previous observations in zebrafish [14]. The same patterns of variation for these genes were also observed in other peripheral organs: muscle (for *g6pd* encoding genes) and intestine (for *fasn*) (table 2).

Gluconeogenesis is the reverse pathway of glycolysis, allowing cells to produce endogenous glucose from pyruvate and gluconeogenic amino acids. This pathway is downregulated in mammals fed with dietary carbohydrates. As for other teleosts fish, gluconeogenesis-related genes are globally not affected by the nutritional status [14,15,39], even though regulation can also occur at enzymatic level. In the present study, gluconeogenesis-related genes transcript levels were not regulated by dietary carbohydrates, similar to previous descriptions in zebrafish, common carp or tilapia [14,40,41]. Of note, we analysed for the first time the *g6pcb2* gene in a non-salmonid species, but we were not able to detect any mRNA in the liver. This gene is a teleost-specific *glucose-6-phosphatase* encoding gene recently discovered in trout and proposed to be involved in its glucose-intolerant phenotype [15,42]. Our result suggests that *g6pcb2* regulation could be species-dependent.

### 3.3.3. Muscle: a major site for glucose disposal

Because of its mass and contractile activity, muscle represents a major captor of glucose for energy purpose. Glut4 is the main glucose transporter in muscle. However, *glut4* mRNA level was not affected in our experimental conditions (table 2). This is in accordance with previous observations in the omnivorous tilapia fed a 30% carbohydrate diet. Concerning glycolysis, only one gene, *pfkmb*, was found to be upregulated by nutritional status even though not affected by the dietary composition. Again this finding is in accordance with the previously published data in omnivorous tilapia fed a high carbohydrate content [17]. As muscle is not recognized as a gluconeogenic organ, we did not assess this pathway in the present study [43].

### 3.3.4. Brain: central glucose and nutrient sensing

The brain is the hub of nutrient sensing, cross-talking with peripheral organs such as intestine, liver and muscle to regulate food intake and metabolism. In fish brain, mostly in rainbow trout (for review, see [44]), the presence of glucosensing mechanisms was demonstrated in the hypothalamus and hindbrain [45–47]. These mechanisms relate to the control of food intake as well as to counter-regulatory responses to changes in glucose circulating levels, and result in enhanced glycolytic potential in response to increased levels of extracellular glucose. More recently, a glucosensing system was highlighted in the telencephalon of rainbow trout [9] and was hypothesized to have a hedonic or reward role (i.e. desire to consume palatable food independently of energy balance [48]) rather than a homeostatic one (i.e. sensing endocrine and metabolic signals informing brain on peripheral energy status [49]) like in the hypothalamus. We thus investigated glucose and lipid metabolism-related genes in these brain regions (table 3).

Except the weak decrease in *glut3a* mRNA level in hypothalamus, the majority of changes observed on glucose metabolism-related genes products occurred in the telencephalon (table 3). Even if unexpectedly downregulated by the nutritional status, the detection of *pfkla* and *pklr* transcripts in the telencephalon signed for a glycolytic potential of the brain in the surface *A. mexicanus*, as previously described in several teleost species regardless of their nutritional preferences [50–52]. Surprisingly, no *gck* mRNA was detected in the brain, although it has been demonstrated to exist in several fish species [53–55]. Moreover, by the detection of *fbp1a* and *pck2* mRNA in the telencephalon, one cannot exclude the presence of gluconeogenesis pathway in the brain of the species, even though the present study may be in the future completed by enzymatic activity analysis. As for glycolytic-related genes, glucose transporters *glut1a*, *glut3a* and *glut8* mRNA levels tended to decrease in fed fish compared to fasted ones, but they were not affected by the composition of the diet. Interestingly, in zebrafish, *glut3* mRNA level was also unchanged after glucose injection in the brain [56]. Of note, *glut2* mRNA was not detected in the brain areas studied here, whereas it is regulated by glucose in the carnivorous rainbow trout [45]. Overall, our results showed that glucose metabolism-related genes were lightly affected in the brain by both nutritional status and dietary composition. It has been previously suggested that some fish brains may not rely on circulating glucose (i.e. exogenous sources) to supply their function, but that glycogen storage may be the proximate glucose source under time-limited metabolic challenge [10]. In that respect, surface *A. mexicanus* has been shown to quickly mobilize glycogen storage at the whole body level during a fasting period [37]. Exploring specifically the glycogen content of its brain could be of interest to better understand their metabolism in future investigations. It should be also noted that our sampling was done after 4 days of feeding but the peak of expression of such markers could have occurred earlier during the first days of re-nutrition as it was previously observed in trout [57]. Nevertheless, like in the liver or intestine, *fasn* mRNA level increased in the hypothalamus and the hindbrain in fed fish compared to fasted fish, indicating the nutritional transition occurred. Unexpectedly, *g6pd* mRNA levels were lower in fed fish than in fasted fish.

In conclusion, in our experimental conditions, the telencephalon seemed to be the most 'responsive' brain area after a change in nutritional status. This result is in accordance with a role in the hedonic and reward system control of food intake of this area proposed to be conserved in teleost [9].

In summary, these results show that mRNA level data tended to discriminate fasted fish from fed fish, but rarely showed an influence of the dietary composition. As previously mentioned, our study was conducted on mature fish and it is well admitted in other species that glucose metabolism could be modified and dietary carbohydrates better metabolized at this stage of life, because of the gametogenesis requirements [58–60]. Finally, it is important to note that we quantified responses of glycolysis-relevant genes to diet at the transcript level. Whether diet invokes differential responses on translation and enzyme kinetics remains to be ascertained and worthy of further study.

**Table 3.** mRNA levels of glucose metabolism-related genes in hypothalamus, hindbrain and telencephalon of fasted fish (F) or fed the no (NC) or high (HC) carbohydrate diet. Data are presented as mean ± s.d. value (hypothalamus: $n = 5$ fish for F and NC, $n = 4$ for HC; hindbrain: $n = 6$ fish for F and HC and $n = 5$ fish for NC; telencephalon: $n = 5$ fish for F and NC, $n = 6$ fish for HC). A Kruskal–Wallis non-parametric test was performed between all conditions ($p$-values are indicated in the table) and when significant ($p < 0.05$, lines in bold) it was followed by a Tukey test as a *post hoc* analysis. Different letters stand for significant differences between conditions. Additional Kruskal–Wallis non-parametric test was performed between NC and HC conditions ($p$-values not shown).

| genes | hypothalamus | | | | hindbrain | | | | telencephalon | | | |
|---|---|---|---|---|---|---|---|---|---|---|---|---|
| | F | NC | HC | *p*-value | F | NC | HC | *p*-value | F | NC | HC | *p*-value |
| *gluconeogenesis* | | | | | | | | | | | | |
| g6pca | | not detected | | | | not detected | | | | not detected | | |
| g6pcb1 | | not detected | | | | not detected | | | | not detected | | |
| g6pcb2 | | not detected | | | | not detected | | | | not detected | | |
| fbp1a | 0.9 ± 0.2 | 1.1 ± 0.2 | 0.9 ± 0.2 | 0.29 | 0.9 ± 0.2 | 1.1 ± 0.3 | 1.2 ± 0.9 | 0.72 | 1.0 ± 0.2 | 0.9 ± 0.3 | 1.0 ± 0.4 | 0.73 |
| fbp1b | | not detected | | | | not detected | | | | not detected | | |
| fbp2 | | not detected | | | | not detected | | | | not detected | | |
| pck1 | | not detected | | | | not detected | | | | not detected | | |
| pck2 | 1.0 ± 0.6 | 1.0 ± 0.2 | 1.1 ± 0.6 | 0.79 | 0.7 ± 0.6 | 1.2 ± 0.3 | 0.9 ± 0.7 | 0.24 | 0.8 ± 0.8 | 1.3 ± 0.3 | 1.1 ± 0.7 | 0.49 |
| *glycolysis* | | | | | | | | | | | | |
| gck | | not detected | | | | not detected | | | | not detected | | |
| pfkla | 1.2 ± 0.1 | 0.9 ± 0.2 | 1.0 ± 0.2 | 0.05 | 0.9 ± 0.2 | 0.9 ± 0.1 | 0.9 ± 0.2 | 0.74 | 1.2 ± 0.2 | 0.9 ± 0.1 | 0.9 ± 0.3 | **0.04** |
| pfklb | 1.7 ± 0.5 | 1.0 ± 0.6 | 0.7 ± 0.5 | 0.05 | 1.1 ± 0.5 | 1.0 ± 0.7 | 0.9 ± 0.7 | 0.63 | 1.6 ± 0.5 | 0.8 ± 0.5 | 0.9 ± 0.7 | 0.09 |
| pklr | 1.3 ± 0.3 | 1.1 ± 0.2 | 1.1 ± 0.2 | 0.53 | 1.0 ± 0.3 | 0.7 ± 0.1 | 1.0 ± 0.9 | 0.13 | 1.3 ± 0.5[a] | 0.7 ± 0.2[b] | 0.8 ± 0.3[a,b] | **0.02** |

(*Continued.*)

**Table 3.** (*Continued.*)

| genes |  |  |  |  | hindbrain |  |  |  | telencephalon |  |  |  |
|---|---|---|---|---|---|---|---|---|---|---|---|---|
|  | F | NC | HC | p-value | F | NC | HC | p-value | F | NC | HC | p-value |
| *transporter* |  |  |  |  |  |  |  |  |  |  |  |  |
| glut1a | $1.1 \pm 0.1$ | $1.0 \pm 0.3$ | $1.0 \pm 0.1$ | 0.78 | $0.2 \pm 0.2$ | $0.6 \pm 0.1$ | $0.7 \pm 0.2$ | 0.05 | $1.6 \pm 0.4^a$ | $0.8 \pm 0.1^b$ | $0.8 \pm 0.2^{a,b}$ | **0.01** |
| glut1b | $1.1 \pm 0.2$ | $1.0 \pm 0.4$ | $0.9 \pm 0.4$ | 0.82 | $0.2 \pm 0.1$ | $0.4 \pm 0.2$ | $0.6 \pm 0.3$ | 0.59 | $1.3 \pm 0.4$ | $0.8 \pm 0.2$ | $1.0 \pm 0.3$ | 0.09 |
| glut2 |  | not detected |  |  |  | not detected |  |  |  | not detected |  |  |
| sglt1a |  | not detected |  |  |  | not detected |  |  |  | not detected |  |  |
| sglt1b |  | not detected |  |  |  | not detected |  |  |  | not detected |  |  |
| glut3a | $1.0 \pm 0.0$ | $0.9 \pm 0.1$ | $0.9 \pm 0.1$ | **0.03** | $0.9 \pm 0.1$ | $1.0 \pm 0.1$ | $1.0 \pm 0.2$ | 0.89 | $1.4 \pm 0.4$ | $1.1 \pm 0.1$ | $1.0 \pm 0.2^{\dagger}$ | **0.03** |
| glut3b | $1.1 \pm 0.4$ | $1.1 \pm 0.2$ | $1.0 \pm 0.1$ | 0.43 | $1.4 \pm 1.2$ | $0.6 \pm 0.1$ | $0.7 \pm 0.4$ | 0.42 | $0.9 \pm 0.6$ | $1.0 \pm 0.8$ | $0.9 \pm 0.4$ | 0.92 |
| glut8 | $1.3 \pm 0.4$ | $0.9 \pm 0.2$ | $1.1 \pm 0.3$ | 0.29 | $1.1 \pm 0.3$ | $0.9 \pm 0.3$ | $0.9 \pm 0.1$ | 0.19 | $1.6 \pm 0.5$ | $0.8 \pm 0.2$ | $0.9 \pm 0.3$ | **0.03** |
| *lipogenesis* |  |  |  |  |  |  |  |  |  |  |  |  |
| g6pda | $1.2 \pm 0.2$ | $0.9 \pm 0.3$ | $0.9 \pm 0.1$ | 0.13 | $0.9 \pm 0.2$ | $0.9 \pm 0.2$ | $0.9 \pm 0.3$ | 0.97 | $1.2 \pm 0.1^a$ | $0.8 \pm 0.3^b$ | $1.0 \pm 0.3^{a,b}$ | **0.04** |
| g6pdb | $1.2 \pm 0.1$ | $0.9 \pm 0.2$ | $1.0 \pm 0.1$ | 0.05 | $0.8 \pm 0.3$ | $0.8 \pm 0.2$ | $0.8 \pm 0.1$ | 0.83 | $1.4 \pm 0.6^a$ | $1.1 \pm 0.1^b$ | $1.0 \pm 0.2^{a,b}$ | **0.04** |
| fasn | $0.7 \pm 0.1^a$ | $1.1 \pm 0.2^b$ | $1.2 \pm 0.2^b$ | **0.01** | $0.8 \pm 0.1^a$ | $1.3 \pm 0.2^b$ | $1.2 \pm 0.2^b$ | **0.004** | $0.8 \pm 0.1$ | $0.9 \pm 0.2$ | $0.9 \pm 0.3$ | 0.19 |

†Significant differences between NC and HC.

# 4. Conclusion

In the present study, we provide a first overview in the surface *A. mexicanus* of the intermediary metabolism response evaluated at the molecular level to nutritional status and to dietary modification of protein-to-carbohydrate content. Our results underlined only few metabolic changes induced by the refeeding, which mostly concerned lipid metabolism. Our findings also highlighted modest metabolic changes in response to changes in protein-to-carbohydrate content, meaning that the surface *A. mexicanus*, at least when sexually mature, can adapt to a nutrition either enriched in digestible carbohydrates and/or containing low protein amount. As for tilapia, surface *A. mexicanus* fed a high carbohydrate diet did not display at transcript level any downregulation of hepatic gluconeogenesis, induction of lipogenesis or muscle glycolysis which could have explained the absence of hyperglycaemia and the glucose tolerance of this species. These data are thus more related to an omnivorous/opportunistic-like metabolism than to a highly carnivorous regime as first described for this species. Experiments increasing the digestible carbohydrate content in the diet and long-term experiments are, however, required to be able to definitively state on the ability of this fish to grow and reproduce on a fish meal-substituted diet. Moreover, the present study focused on sexually mature adult fish and investigation at other stages is needed to establish a proper nutritional table for the whole life cycle of the species and to conduct further nutritional/metabolic experiment on this model.

Ethics. Investigations were conducted according to the guiding principles for the use and care of laboratory animals and in compliance with French and European regulations on animal welfare (Décret 2001-464, 29 May 2001, and Directive 2010/63/EU, respectively). S.R.'s authorization for use of animals in research including *Astyanax mexicanus* is 91-116. The Paris Centre-Sud Ethic Committee approved the protocol authorization number 2017-04#8545 related to the present research.

Data accessibility. Remaining biological material used here (mRNA and proteins) is available at the NuMeA lab, INRA. Data used in the present study are available in electronic supplementary material, table S2.

Authors' contributions. L.M. and S.R. designed the study. L.M. wrote the manuscript. S.R. managed the experiment with the help of S.Pè. L.M., M.M., T.C. and S.R. sampled the fish. S.Pè. and L.V. bred and fed the fish and helped during the sampling. M.M. and E.P.-J. performed molecular analysis. K.D. performed western blot analysis. F.T. elaborated the diets. S.Pa. participated in the design of the study and the writing of the manuscript. All authors read and corrected the manuscript.

Competing interests. The authors declare that there are no conflicts/competing of interests.

Funding. L.M. received financial support from the INRA PHASE department to conduct the present research. Experiments in S.R. laboratory were supported by CNRS and an Equipe FRM grant.

Acknowledgements. We would like to thank Anne Surget for diet composition analyses.

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
