## [Reviewer comments · Royal Society Open Science]

Review History

RSOS-191853.R0 (Original submission)

Review form: Reviewer 1 (Sergio Polakof)

Is the manuscript scientifically sound in its present form?

Yes

Are the interpretations and conclusions justified by the results?

Yes

Is the language acceptable?

Yes

Do you have any ethical concerns with this paper?

No

Have you any concerns about statistical analyses in this paper?

No

Recommendation?

Accept with minor revision (please list in comments)

Comments to the Author(s)

The study by Marandel et al explores at the molecular level the intermediary metabolism in the surface Mexican fish.

Two main nutritional tests were used to this aim: first, the fasted-to-feeding transition, then the comparison between a low and a high carbohydrate diet.

The study seems to be well conducted in the light of the proposed hypotheses.

Methods are correct and adapted.

However, several suggestions could be considered to improve the readability of the manuscript and the discussion accuracy. Some of them can be found below:

L2: may be mention why this species is important?

L4/L306: please, consider to modify as "evaluate at the molecular level the intermediary metabolism..."

L7: here and then in the main text, hyperglycemia should be defined. Hyperglycaemia is often considered as high blood glucose levels, which is the case in the present study. Then, the authors should explain why no hyperglycemia is observed.

In the introduction is not clear what the problem is: it seems that this fish is used as a lab model, but not necessarily for nutrition? Then why it would be interesting to establish the nutritional requirements? Is there any perspective to use this animal for nutrition studies? In the light of the presented results, the picture is not very clear.

L75: the age of animals seems highly variable. Even for adult animals, this should be taken into account in the discussion.

L88: glucose has been measured in the blood, not in the plasma. Actually, to verify the blood levels with plasma or serum assessments is highly recommended. Glucometer values are validated for humans, but not necessarily fish.

Blood glucose levels are extremely low, particularly for omnivorous fish. May be the fasting period was too long ? Usually salmonids are highly adapted to fasting due to its evolutionary adaptation to migration. But this could not be the case for the Mexican species. If those fish were actually hypoglycaemic, then their response to re-feeding could be reduced or blunted. This could also explain the low fed/fasting ration for glycaemia.

Overall, the mRNA levels analyses showed little changes to the meal and in particular to the meal composition. Although is vaguely mentioned, the authors should further discuss the fact that those results are only applied to the molecular regulation of the metabolism. More functional data should be provided in future studies to expand the knowledge of the responsiveness of the metabolism.

Review form: Reviewer 2 (Nicolas Rohner)

Is the manuscript scientifically sound in its present form?

Yes

Are the interpretations and conclusions justified by the results?

Yes

Is the language acceptable?

Yes

Do you have any ethical concerns with this paper?

No

Have you any concerns about statistical analyses in this paper?

No

Recommendation?

Accept as is

Comments to the Author(s)

The manuscript: "Nutritional regulation of glucose-metabolism related genes in the emerging teleost model surface Mexican fish: a first exploration" by Marandel et al. studies the expression of glucose metabolism genes in response to diets with two different carbohydrate contents (medium-high and low) in *Astyanax mexicanus*. The authors focused on the surface population of this species. While in the long term it may be interesting to extend this analysis to the cave populations as well, the study provides an important foundation to understand dietary requirements in these fish. The authors nicely dissect the glucose metabolism in the gut, liver, muscle and brain of these fish. Contrary to some previous beliefs the authors provide evidence that these fish are more omnivorous than carnivorous. This is in line with the cavefish populations being more opportunistic eaters as well and will be important information to have for husbandry and future metabolism studies in this species. I have only one minor comment:

The title: surface Mexican fish is not the correct nomenclature. I suggest replacing "surface Mexican fish" by "*Astyanax mexicanus*" or "Mexican Tetra surface fish"

Decision letter (RSOS-191853.R0)

22-Jan-2020

Dear Dr Marandel

On behalf of the Editors, I am pleased to inform you that your Manuscript RSOS-191853 entitled "Nutritional regulation of glucose-metabolism related genes in the emerging teleost model surface Mexican fish: a first exploration" has been accepted for publication in Royal Society Open Science subject to minor revision in accordance with the referee suggestions. Please find the referees' comments at the end of this email.

The reviewers and handling editors have recommended publication, but also suggest some minor revisions to your manuscript. Therefore, I invite you to respond to the comments and revise your manuscript.

- Ethics statement

- Data accessibility

It is a condition of publication that all supporting data are made available either as supplementary information or preferably in a suitable permanent repository. The data accessibility section should state where the article's supporting data can be accessed. This section should also include details, where possible of where to access other relevant research materials such as statistical tools, protocols, software etc can be accessed. If the data has been deposited in an external repository this section should list the database, accession number and link to the DOI

for all data from the article that has been made publicly available. Data sets that have been deposited in an external repository and have a DOI should also be appropriately cited in the manuscript and included in the reference list.

If you wish to submit your supporting data or code to Dryad (<http://datadryad.org/>), or modify your current submission to dryad, please use the following link:
<http://datadryad.org/submit?journalID=RSOS&manu=RSOS-191853>

- **Competing interests**

- **Authors' contributions**

- **Acknowledgements**

- **Funding statement**

Because the schedule for publication is very tight, it is a condition of publication that you submit the revised version of your manuscript before 31-Jan-2020. Please note that the revision deadline will expire at 00.00am on this date. If you do not think you will be able to meet this date please let me know immediately.

When submitting your revised manuscript, you will be able to respond to the comments made by the referees and upload a file "Response to Referees" in "Section 6 - File Upload". You can use this to document any changes you make to the original manuscript. In order to expedite the

processing of the revised manuscript, please be as specific as possible in your response to the referees. We strongly recommend uploading two versions of your revised manuscript:

If your manuscript is newly submitted and subsequently accepted for publication, you will be asked to pay the article processing charge, unless you request a waiver and this is approved by Royal Society Publishing. You can find out more about the charges at <https://royalsocietypublishing.org/rsos/charges>. Should you have any queries, please contact openscience@royalsociety.org.

Kind regards,
Lianne Parkhouse
Royal Society Open Science
openscience@royalsociety.org

on behalf of Dr Punidan Jeyasingh (Associate Editor) and Kevin Padian (Subject Editor)
openscience@royalsociety.org

Associate Editor Comments to Author (Dr Punidan Jeyasingh):

This manuscript reports data on differential expression of glucose metabolism genes due to diet in the surface dwelling Mexican tetra. The manuscript was assessed by two experts. Both experts were highly positive about the work. They make suggestions to improve the paper. I felt the comments were fair and constructive. With much gratitude to the experts, I invite the authors to make these adjustments, after which it should be ready for press.

Reviewer comments to Author:

Reviewer: 1

Comments to the Author(s)

The study by Marandel et al explores at the molecular level the intermediary metabolism in the surface Mexican fish.

Two main nutritional tests were used to this aim: first, the fasted-to-feeding transition, then the comparison between a low and a high carbohydrate diet.

The study seems to be well conducted in the light of the proposed hypotheses.

Methods are correct and adapted.

However, several suggestions could be considered to improve the readability of the manuscript and the discussion accuracy. Some of them can be found below:

L2: may be mention why this species is important?

L4/L306: please, consider to modify as "evaluate at the molecular level the intermediary metabolism..."

L7: here and then in the main text, hyperglycemia should be defined. Hyperglycaemia is often considered as high blood glucose levels, which is the case in the present study. Then, the authors should explain why no hyperglycemia is observed.

In the introduction is not clear what the problem is: it seems that this fish is used as a lab model, but not necessarily for nutrition? Then why it would be interesting to establish the nutritional requirements? Is there any perspective to use this animal for nutrition studies? In the light of the presented results, the picture is not very clear.

L75: the age of animals seems highly variable. Even for adult animals, this should be taken into account in the discussion.

L88: glucose has been measured in the blood, not in the plasma. Actually, to verify the blood levels with plasma or serum assessments is highly recommended. Glucometer values are validated for humans, but not necessarily fish.

Blood glucose levels are extremely low, particularly for omnivorous fish. May be the fasting period was too long? Usually salmonids are highly adapted to fasting due to its evolutionary adaptation to migration. But this could not be the case for the Mexican species. If those fish were actually hypoglycaemic, then their response to re-feeding could be reduced or blunted. This could also explain the low fed/fasting ration for glycaemia.

Overall, the mRNA levels analyses showed little changes to the meal and in particular to the meal composition. Although is vaguely mentioned, the authors should further discuss the fact that those results are only applied to the molecular regulation of the metabolism. More functional

data should be provided in future studies to expand the knowledge of the responsiveness of the metabolism.

Reviewer: 2

Comments to the Author(s)

The manuscript: "Nutritional regulation of glucose-metabolism related genes in the emerging teleost model surface Mexican fish: a first exploration" by Marandel et al. studies the expression of glucose metabolism genes in response to diets with two different carbohydrate contents (medium-high and low) in *Astyanax mexicanus*. The authors focused on the surface population of this species. While in the long term it may be interesting to extend this analysis to the cave populations as well, the study provides an important foundation to understand dietary requirements in these fish. The authors nicely dissect the glucose metabolism in the gut, liver, muscle and brain of these fish. Contrary to some previous beliefs the authors provide evidence that these fish are more omnivorous than carnivorous. This is in line with the cavefish populations being more opportunistic eaters as well and will be important information to have for husbandry and future metabolism studies in this species. I have only one minor comment:

The title: surface Mexican fish is not the correct nomenclature. I suggest replacing "surface Mexican fish" by "*Astyanax mexicanus*" or "Mexican Tetra surface fish"

Author's Response to Decision Letter for (RSOS-191853.R0)

See Appendix A.

Decision letter (RSOS-191853.R1)

27-Jan-2020

Dear Dr Marandel:

On behalf of the Editors, I am pleased to inform you that your Manuscript RSOS-191853.R1 entitled "Nutritional regulation of glucose-metabolism related genes in the emerging teleost model Mexican Tetra surface fish: a first exploration" has been accepted for publication in Royal Society Open Science subject to minor revision in accordance with the referee suggestions. Please find the referees' comments at the end of this email.

The reviewers and Subject Editor have recommended publication, but also suggest some minor revisions to your manuscript. Therefore, I invite you to respond to the comments and revise your manuscript.

- Ethics statement

- Data accessibility

<http://datadryad.org/submit?journalID=RSOS&manu=RSOS-191853.R1>

- Competing interests

- Authors' contributions

- Acknowledgements

- Funding statement

Because the schedule for publication is very tight, it is a condition of publication that you submit the revised version of your manuscript before 05-Feb-2020. Please note that the revision deadline will expire at 00.00am on this date. If you do not think you will be able to meet this date please let me know immediately.

To revise your manuscript, log into <https://mc.manuscriptcentral.com/rsos> and enter your Author Centre, where you will find your manuscript title listed under "Manuscripts with Decisions". Under "Actions," click on "Create a Revision." You will be unable to make your

revisions on the originally submitted version of the manuscript. Instead, revise your manuscript and upload a new version through your Author Centre.

on behalf of Dr Punidan Jeyasingh (Associate Editor) and Kevin Padian (Subject Editor)
openscience@royalsociety.org

Associate Editor Comments to Author (Dr Punidan Jeyasingh):

The authors have addressed all reviewer concerns. The manuscript is improved, and almost ready for press. I would like to see the wording in L309 revised. I see that the authors included this in response to reviewer comment. However, is it not very clear. I suggest something like this: "Finally, it is important to note that we quantified responses of glycolysis relevant genes to diet at the transcript level. Whether diet invokes differential responses on translation and enzyme kinetics remains to be ascertained and worthy of further study."

Author's Response to Decision Letter for (RSOS-191853.R1)

See Appendix B.

Decision letter (RSOS-191853.R2)

30-Jan-2020

Dear Dr Marandel,

It is a pleasure to accept your manuscript entitled "Nutritional regulation of glucose-metabolism related genes in the emerging teleost model Mexican Tetra surface fish: a first exploration" in its current form for publication in Royal Society Open Science.

Kind regards,
Lianne Parkhouse
Royal Society Open Science
openscience@royalsociety.org

on behalf of Dr Punidan Jeyasingh (Associate Editor) and Kevin Padian (Subject Editor)
openscience@royalsociety.org

Appendix A

Reviewer: 1

Comments to the Author(s)

The study by Marandel et al explores at the molecular level the intermediary metabolism in the surface Mexican fish. Two main nutritional tests were used to this aim: first, the fasted-to-feeding transition, then the comparison between a low and a high carbohydrate diet. The study seems to be well conducted in the light of the proposed hypotheses. Methods are correct and adapted. However, several suggestions could be considered to improve the readability of the manuscript and the discussion accuracy. Some of them can be found below:

L2: may be mention why this species is important?

A clarification has been added.

L4/L306: please, consider to modify as “evaluate at the molecular level the intermediary metabolism...”

It is still mentioned L8 and we added it L306.

L7: here and then in the main text, hyperglycemia should be defined. Hyperglycaemia is often considered as high blood glucose levels, which is the case in the present study. Then, the authors should explain why no hyperglycemia is observed.

As the summary is limited to 200 words, we added this information in the main text (lines 178-179).

In the introduction is not clear what the problem is: it seems that this fish is used as a lab model, but not necessarily for nutrition? Then why it would be interesting to establish the nutritional requirements? Is there any perspective to use this animal for nutrition studies? In the light of the presented results, the picture is not very clear.

The model is not use for nutritional study *per se* for the moment. However and regarding the phenotype of the cave form (hyperphagia (elevated appetite) to increase food consumption, fat deposition and starvation resistance: see Xiong S et al, 2018 Dev. Biol.), it is a realistic perspective to also consider the surface form in an evolutionary aim for nutritional/metabolic related studies.

It is in addition really important to have information about nutrition/metabolism to be able to correctly breed a lab animal in general as growth and nutrition strongly interact with health and reproduction for instance.

L75: the age of animals seems highly variable. Even for adult animals, this should be taken into account in the discussion.

We agree with this point. Fish were not tagged, their age can only be revealed in our experiment by their length/weight as mentioned l77. We performed correlation analysis between the length/weight of the fish and mRNA levels, and never found significant correlation suggesting that the age of the fish did not influence our results. We added this information l192.

L88: glucose has been measured in the blood, not in the plasma. Actually, to verify the blood levels with plasma or serum assessments is highly recommended. Glucometer values are validated for humans, but not necessarily fish.

We agree, however fish were too small to get enough plasma to measure glycaemia by a classical colorimetric kit. Importantly, our glucometer was the same model as the one used in a previous study on this species (Riddle et al, Nature, 2018) and we get comparable results. We corrected “plasma” by “blood”.

Blood glucose levels are extremely low, particularly for omnivorous fish. May be the fasting period was too long ? Usually salmonids are highly adapted to fasting due to its evolutionary adaptation to migration. But this could not be the case for the Mexican species. If those fish were actually hypoglycaemic, then their response to re-feeding could be reduced or blunted. This could also explain the low fed/fasting ration for glycaemia.

This could be. However, the cave form is adapted to long period of starvation. Four days of fasting is not considered as a long one in nutrition but a shorter one could be tested in the future on this species.

Overall, the mRNA levels analyses showed little changes to the meal and in particular to the meal composition. Although is vaguely mentioned, the authors should further discuss the fact that those results are only applied to the molecular regulation of the metabolism. More functional data should be provided in future studies to expand the knowledge of the responsiveness of the metabolism.

Such perspective is mentioned I233 and I249. We added a sentence I309 to reinforce this idea.

Reviewer: 2

The manuscript: “Nutritional regulation of glucose-metabolism related genes in the emerging teleost model surface Mexican fish: a first exploration” by Marandel et al. studies the expression of glucose metabolism genes in response to diets with two different carbohydrate contents (medium-high and low) in *Astyanax mexicanus*. The authors focused on the surface population of this species. While in the long term it may be interesting to extend this analysis to the cave populations as well, the study provides an important foundation to understand dietary requirements in these fish. The authors nicely dissect the glucose metabolism in the gut, liver, muscle and brain of these fish. Contrary to some previous beliefs the authors provide evidence that these fish are more omnivorous than carnivorous. This is in line with the cavefish populations being more opportunistic eaters as well and will be important information to have for husbandry and future metabolism studies in this species. I have only one minor comment:

The title: surface Mexican fish is not the correct nomenclature. I suggest replacing “surface Mexican fish” by “*Astyanax mexicanus*” or “Mexican Tetra surface fish”

We modified the title according to the reviewer suggestion.

Appendix B

St Pée-sur-Nivelle, January, 27th, 2020

Dear Editor,

Please find enclosed the corrected manuscript entitled “Nutritional regulation of glucose-metabolism related genes in the emerging teleost model Mexican Tetra surface fish: a first exploration” by Marandel *et al.* for acceptance to publication in Royal Society Open Science.

We answered all the reviewers’ comments and corrected the manuscript accordingly. We corrected 1309 accroding to the editor’remark.

We hope you will find our manuscript suitable for publication in Royal Society Open Science.

Sincerely,

Dr Lucie MARANDEL
